# Molecular and Pathological Characterization of Classical Swine Fever Virus Genotype 2 Strains Responsible for the 2013–2018 Outbreak in Colombia

**DOI:** 10.3390/v15122308

**Published:** 2023-11-24

**Authors:** Erin Robert, Kalhari Goonewardene, Lindsey Lamboo, Orlando Perez, Melissa Goolia, Charles Lewis, Cassidy N. G. Erdelyan, Oliver Lung, Katherine Handel, Estella Moffat, Carissa Embury-Hyatt, Nancy Naranjo Amaya, Claudia Patricia Calderón Parra, Diana Cristina Gómez Rueda, Maria Antonia Rincón Monroy, Alfonso Clavijo, Aruna Ambagala

**Affiliations:** 1Canadian Food Inspection Agency, National Centre for Foreign Animal Disease, Winnipeg, MB R3E 3R2, Canada; erin.robert@inspection.gc.ca (E.R.); kalhari.goonewardene@inspection.gc.ca (K.G.); lindsey.lamboo@phac-aspc.gc.ca (L.L.); orlando.perez@phac-aspc.gc.ca (O.P.); melissa.goolia@inspection.gc.ca (M.G.); charles.lewis@usda.gov (C.L.); cass.erdelyan@inspection.gc.ca (C.N.G.E.); oliver.lung@inspection.gc.ca (O.L.); katherine.handel@inspection.gc.ca (K.H.); estella.moffat@inspection.gc.ca (E.M.); carissa.emburyhyatt@inspection.gc.ca (C.E.-H.); alfonso.clavijo@usda.gov (A.C.); 2National Veterinary Laboratory, Instituto Colombiano Agropecurio, Bogota 110911, DC, Colombia; nancy.naranjo@ica.gov.co (N.N.A.); claudia.calderon@ica.gov.co (C.P.C.P.); diana.gomez@ica.gov.co (D.C.G.R.); maria.rincon@ica.gov.co (M.A.R.M.); 3Department of Animal Science, University of Manitoba, Winnipeg, MB R3T 2N2, Canada; 4Department of Comparative Biology and Experimental Medicine, Faculty of Veterinary Medicine, University of Calgary, Calgary, AB T2N 1N4, Canada

**Keywords:** classical swine fever, pathogenesis, genotype, moderately virulent, Colombia

## Abstract

Classical swine fever (CSF) is a highly contagious transboundary viral disease of domestic and wild pigs. Despite mass vaccination and continuous eradication programs, CSF remains endemic in Asia, some countries in Europe, the Caribbean and South America. Since June 2013, Northern Colombia has reported 137 CSF outbreaks, mostly in backyard production systems with low vaccination coverage. The purpose of this study was to characterize the virus responsible for the outbreak. Phylogenetic analysis based on the full-length E2 sequence shows that the virus is closely related to CSF virus (CSFV) genotype 2.6 strains circulating in Southeast Asia. The pathotyping experiment suggests that the virus responsible is a moderately virulent strain. The 190 nucleotide stretch of the E2 hypervariable region of these isolates also shows high similarity to the CSFV isolates from Colombia in 2005 and 2006, suggesting a common origin for the CSF outbreaks caused by genotype 2.6 strains. The emergence of genotype 2.6 in Colombia suggests a potential transboundary spread of CSFV from Asia to the Americas, complicating the ongoing CSF eradication efforts in the Americas, and emphasizes the need for continuous surveillance in the region.

## 1. Introduction

Classical swine fever (CSF) remains one of the most important transboundary viral diseases of domestic and wild pigs [1]. It has a tremendous impact on the pig industry and is therefore notifiable to the World Organization for Animal Health (WOAH). WOAH recognizes most countries from Europe and Oceania, Kazakhstan, the USA, Canada, Mexico, Costa Rica, Argentina, Chile, Paraguay, Uruguay and some areas of Brazil and Colombia as free of CSF [2]. Classical swine fever virus (CSFV), the causative agent of CSF, is a positive-sense single-stranded RNA virus belonging to the genus *Pestivirus* of the family *Flaviviridae* [1]. Other members of the genus *Pestivirus* include border disease virus (BDV) and bovine viral diarrhea virus (BVDV). In contrast to BVDV and BDV, which infect a wide range of ruminant species and pigs, CSFV only infects pigs. The CSFV genome is approximately 12.3 kb and it consists of a single open reading frame (ORF) that encodes a polyprotein of 3898 amino acids [1]. All CSFV strains belong to a single serotype but have been divided into three main genotypes (one to three), and numerous sub-genotypes. Historically, genotype 1 has been divided into six sub-genotypes (1.1 to 1.6), and genotype 2 into three sub-genotypes (2.1 to 2.3). Recently a new classification system for CSFV has been proposed, dividing CSFV strains into five genotypes (one to five), and the genotype 2 strains into seven sub-genotypes (2.1 to 2.7) [3]. The two new genotypes (4 and 5) comprise of the distantly related CSFV strain “congenital tremor” (Great Britain/1964) and two strains from Korea (KR/1998, KR/1999), respectively. 

Based on virulence, CSFV strains have been divided into high-, moderate-, and low-virulence viruses [1]. There is no clear correlation established between CSFV genotypes and virulence. However, the most virulent strains are found in genotype 1, and generally genotype 2 strains are moderately virulent. Genotype 2 strains have been the most prevalent genotype detected across the globe over the last few decades. Clinical signs and mortality in CSF is highly variable and can vary due to host factors such as age, health, concomitant infections, and the immune status of the affected animal, as well as the virulence of the CSFV strain involved [1,4]. The clinical forms of CSF include acute, subacute, chronic, and subclinical. 

Acute CSF is mostly caused by highly virulent CSFV strains, and by moderately virulent CSFV strains when they infect young pigs (weaners) [5,6]. The incubation period is often shorter (3–7 days), with mortality seen within 2 weeks [7]. The pigs develop a high fever (>41 °C), huddle together in groups, and show a loss of appetite, weakness, conjunctivitis, and constipation followed by diarrhea, vomiting, and unsteady gait. Due to central nervous system infection, some pigs show depression, uncoordinated movements and sometimes seizures. Several days after the onset of clinical signs, cyanotic skin lesions appear, particularly in the areas of the ears, abdomen, inner thighs, or extremities. Morbidity and mortality in acute CSF can approach 100% within 10–30 days. Pigs that develop the subacute form of CSF show similar clinical signs, although less severely, and some animals might recover as they develop robust humoral immunity characterized by neutralizing antibodies. Chronic CSF is caused by low-virulent CSFV strains, where the pigs develop mild clinical signs such as intermittent fever, chronic enteritis, respiratory signs, wasting, and diffuse dermatitis [8]. Pigs that develop the chronic form of CSF can survive 2–3 months; this is seen mostly in CSF-endemic countries. Due to virus-induced immunosuppression, CSFV-infected pigs develop secondary bacterial infections (pneumonia, enteritis) and they shed CSFV continuously or intermittently for months, representing a constant source of infection. Subclinical CSF forms are observed in persistently infected piglets because of trans-placental transmission of the virus or postnatal infection of piglets within their first three weeks of life [9,10,11]. These piglets are immune-tolerant to the virus, shed large amounts of CSFV, and after several months develop anorexia, depression, dermatitis, diarrhea, poor growth, and paresis, and eventually succumb to the infection [9,10,11,12]. 

The global distribution of CSFV genotypes shows a distinct geographical pattern [1,13,14]. The isolates belonging to genotype 3 seem to occur solely in Asia [15]. All European CSFV isolates before the 1990s belonged to genotype 1. The CSFV genotypes discovered after 1990 in Europe belonged to genotype 2 [16,17,18]. CSFV genotype 1 and 2 strains appear to co-circulate in India and China [19,20,21,22,23]. Until recently, field isolates from the American continent exclusively belonged to genotype 1, with 1.1 strains from Argentina, Brazil, Colombia, Peru, and Mexico; 1.3 strains from Honduras and Guatemala; 1.4 strains from Cuba; and 1.5 and 1.6 strains from Brazil [24,25,26,27,28,29,30]. For the first time in the Americas, CSFV genotype 2 strains were reported from domestic pigs in samples collected between 2005 and 2006 from La Guajira and Norte de Santander, two northern departments of Colombia [31]. The outbreak was successfully controlled by June 2007, and four years later, northern Colombia was hit by another CSF outbreak [31]. The index case was reported in June 2013 from the municipality Urumita, in the Colombian Department of La Guajira. The outbreak spread to neighboring departments, causing 134 outbreaks, and lasted for five years. Between 2013 and 2018, 134 CSF outbreaks affected the northern part of Colombia, with almost half of them (64; 47%) occurring in 2015. Here, we describe the molecular and pathological characterization of the CSFV strain responsible for the 2013–2018 outbreak.

## 2. Materials and Methods

### 2.1. Field Samples from Colombia

A total of 13 serum (from live pigs) and 10 tissue samples (from dead pigs) collected between June 2013 and September 2017, from 22 pigs (at different ages and mixed breeds) in northern Colombia, were analyzed in this study (Figure 1 and Table 1). All except one sample (#2017-356) were collected from backyard operations. The first two samples (serum and tonsil) were collected from the index case, a backyard pig, on 1 June 2013, in Urumita, a small town 50 km west of the Venezuelan border. 

### 2.2. Nucleic Acid Extraction and RRT-PCR

Total nucleic acid was extracted from the serum and 10% tissue suspensions in Dulbecco’s Phosphate Buffered Saline (DPBS, Corning, Manassas, VA, USA) using the MagMAX™ Pathogen RNA/DNA Kit (Life Technologies, Burlington, ON, Canada) and the MagMAX Express-96 Magnetic Particle Processor (Life Technologies) following the manufacturer’s protocol. Prior to extraction, the tissue samples were homogenized using a Precellys^®^ 24 Tissue Homogenizer (Bertin Technologies, Rockville, MD, USA), as recommended by the manufacturer. Briefly, the tissue samples were added to the Precellys^®^ hard tissue homogenizing tubes CK28 (Bertin Technologies) at a final dilution of 10% in a solution of sterile DPBS (Corning). Samples were subjected to 2 cycles of homogenization. Each homogenizing cycle comprised three 5000 rpm 10 s runs with 10 s pauses. Between the two cycles, samples were chilled on ice for 1 min. Then, the samples were centrifuged at 2000× *g* for 20 min at 4 °C, and the supernatant was then transferred into a sterile 2 mL cryovial placed at −70 °C until extraction. For ear notches and skin biopsies, due to the tougher nature of the tissue, 3 homogenization cycles were carried out, unlike with other tissues. Fifty-five microliters of serum or clarified 10% tissue homogenate was used for the nucleic acid extraction. For buccal swabs, oropharyngeal swabs, and nasal swabs, 300 µL of each sample was used. The amount of CSFV genome in the extracted nucleic acid samples was determined using a previously described real-time RT-PCR assay that targets the 5′ UTR of the CSFV genome [32]. The cycling conditions of the assay were modified to match the protocol of the utilized TaqMan™ Fast Virus 1-Step Master Mix (Thermo Fisher Scientific, Burlington, ON, CA). 

### 2.3. Virus Isolation and Titration

Samples that had Ct values below 30 were subjected to virus isolation using the National Centre for Foreign Animal Diseases’ (NCFAD) virus isolation standard operating procedure (SOP). Briefly, PK-15 cells (Cedarlane, Burlington, ON, Canada) grown to 50–70% confluence in T25 cm^2^ flasks and were inoculated, each with 0.5 mL of serum or 10% clarified tissue suspension sample containing CSFV per flask. The flasks were incubated on a rocker for one hour at 37 °C in a 5% CO_2_ incubator. Afterwards, 4.5 mL of Alpha Minimal Essential Medium (Corning) supplemented with 1% Glutamax, 1% Gentamicin, and 2% irradiated horse serum (both from Thermo Fisher Scientific) was added to each flask, and incubated for an additional 3 days at 5% CO_2_ at 37 °C. Then, the flasks were immediately frozen for a minimum of 2 h at −70 °C. The flasks were thawed, the attached CSFV-infected PK-15 cell monolayer was scraped, and the resulting cell suspension was clarified by centrifugation at 2000× *g* for 20 min. The clarified supernatant was titrated on PK-15 cells to determine the TCID_50_/mL. 

### 2.4. Whole Genome Sequencing

A total of seven isolates were subjected to whole genome sequencing using the Ion S5™ system (Thermo Fisher Scientific). Briefly, total nucleic acids were extracted from the CSFV-infected PK-15 cell supernatant using the MagMAX™ Pathogen RNA/DNA Kit (Thermo Fisher Scientific) and were quantified using the Qubit RNA HS Assay Kit (Catalogue: Q32852, Thermo Fisher Scientific). Samples were then diluted to 10 ng in a final volume of 12 µL and cDNA was generated using SuperScript™ IV VILO™ Master Mix (Thermo Fisher Scientific) as per the instructions in the AmpliSeq™ DL8 (Thermo Fisher Scientific) protocol using a custom AmpliSeq primer panel. Amplification settings were selected based on the number of primers, as per the manufacturer’s protocol (approximately 4600 primers per pool, and therefore 14 amplification cycles with a 16 min anneal/extension time). After amplification on the Ion Chef (Thermo Fisher Scientific), the library was diluted 1:1 in water and loaded back onto the instrument with the required consumables for automated templating and sequencing flow-cell loading. An Ion 530 chip (Thermo Fisher Scientific) with the final library pool loaded on was transferred from the Ion Chef to the Ion S5 system (Thermo Fisher Scientific) for sequencing. 

Preliminary read mapping and consensus sequence generation were performed by running all samples through the CFIA-NCFAD/nf-ionampliseq (v1.0.1) [33] Nextflow [34] pipeline. First, BAM files were to converted to FASTQ with Samtools (v1.11) [35]. Mash screen (v2.2.2) [36] was used to determine the top reference genome from the 97 references included in the original AmpliSeq panel design. The Thermo Fisher TMAP read mapper (v5.12.28) [37] was used to map reads to the top reference genome. Variant calling and AmpliSeq primer trimming was performed with TVC (v5.12.28) [38]. Variant calling results were normalized and filtered with Bcftools (v1.11) [35]. Depth-masked majority consensus sequences were generated from the TMAP read alignments and Bcftools filtered TVC variant calling results. Additional read mapping was performed on more recently published references. First, reads were trimmed 15 bases on both ends and quality filtered with fastp (v0.20.1) [38] using default settings. Consensus sequences were searched against the NCBI nr/nt database using NCBI web BLAST [39,40] (performed between 24 April and 9 May 2023) to determine the closest matching complete reference sequence for each sample. For each sample, the fastp filtered reads were mapped to the top reference genome in Geneious Prime (v2023.0.1, Biomatters, Inc., Boston, MA, USA) [41] using the Minimap2 (v2.24) [42] assembler. A 75% majority consensus sequence was called with a minimum coverage threshold of 10×. This consensus sequence was aligned with the nf-ionampliseq consensus sequence using MAFFT (v7.490) [43] to manually check variants identified by Geneious and TVC. The Geneious consensus sequences were also checked for the presence of a complete coding sequence for the polyprotein gene in Geneious Prime (v2023.0.1) [41] using the Find ORFs tool.

### 2.5. Phylogenetic Analysis

For phylogenetic analysis, full length E2 CSFV sequences were retrieved from GenBank (https://www.ncbi.nlm.nih.gov/genbank/ ((accessed on 22 March 2023), National Library of Medicine, Bethesda, MD, USA) and the CSFV database (http://viro60.tiho-hannover.de/eg/csf/ ((accessed on 22 March 2023) University of Veterinary Medicine, Hannover, Germany). Unique full-length sequences were screened for recombinants using RDP v5.34 [44] with default parameters. Non-recombinant sequences were used to generate a maximum-likelihood (ML) tree to infer phylogeny using IQ-TREE (v2.1.4-beta) [45] with ModelFinder using 1000 bootstrap replicates of the SH-like approximate likelihood ratio test (SH-aLRT). CSFV Congenital Tremor (GenBank: JQ411575) was used as the outgroup. SYM + R4 was the best-fit nucleotide substation model as chosen according to BIC. Country of strain isolation was used as a discrete trait for TreeTime (v0.11.1) mugration model inference [46]. The resulting phylogenetic tree was visualized using ggtree [47] and genotypes were labeled as proposed by Rios et al., 2017 [3] and further discussed by Ganges et al., 2020 [4].

### 2.6. Animal Experiment

To determine the pathogenicity of the CSFV responsible for the 2013–2018 outbreak in Colombia, an animal experiment was conducted using five 6-week old weaned piglets in the biosafety level 3 animal pens at the National Centre for Foreign Animal Disease (NCFAD), Winnipeg. The piglets, Large White x Landrace x Duroc cross, were purchased from a local supplier in Manitoba, housed in a single animal pen and provided feed twice a day, water ad libitum. The piglets were also provided with a heat lamp as a supplementary heat source and toys for environmental enrichment. After seven days of acclimatization, each piglet was infected intra-nasally with CSFV 2016/Pinillos at a dose of 1 × 10^5^ TCID_50_ in 1.0 mL per pig (0.5 mL of virus suspension per nostril). After inoculation, pigs were observed a minimum of twice a day, in the morning and afternoon, and with greater frequency as the animals developed clinical signs. A clinical scoring system [6] was used to record daily clinical findings of each animal, along with their rectal temperatures. Clinical signs detectable were evaluated and scored on a 0–3 point scale according to the severity of the lesion: score 0 = normal; score 1 = slightly altered; score 2 = distinct clinical signs; and score 3 = severe. The clinical signs evaluated included liveliness, body tension, body shape, breathing, walking (neurological signs), skin, eyes (conjunctivitis), appetite, and defecation. When this clinical scoring system is used in conjunction with fever, highly virulent strains give average daily scores >15 with high fever (≥41 °C), moderately virulent strains score between 5 and 15, with moderate fever (40–41 °C), and low-virulent strains score around 2, and low to no fever (≤40 °C).

The day prior to virus inoculation (−1 dpi), and on days 5, 9, 12, 15, 18, 21, and 22 post-inoculation, buccal swabs, pharyngeal swabs, nasal swabs, whole blood, and serum samples were collected from each pig under gaseous anesthesia. On each sampling day, a total of 17 mL of blood was collected from each pig from the jugular vein with a 20G Vacutainer^®^ needle (8 mL in serum separator tubes, 6 mL in EDTA tubes, and 3 mL in sodium citrate tubes, all from Thermo Fisher Scientific). The serum separator tubes were spun at 3000× *g* for 15 min for clarification before being aliquoted and stored. The nasal, buccal, and pharyngeal samples were collected using sterile Puritan™ cotton swabs (Puritan Medical Products, Guilford, ME, USA), each in 1.0 mL of DPBS (Corning). Virus inoculation and sampling were carried out under gaseous anesthesia (2% isoflurane delivered in 100% oxygen), and all procedures involving animals were compliant with the Canadian Council for Animal Care (CCAC) guidelines. The use of animals was approved by the Animal Care Committee at the Canadian Science Centre for Human and Animal Health (CSCHAH) under the animal user document # C-20-001.

Pigs that reached humane end points were euthanized using sodium pentobarbital (240 mg/mL) injected intravenously. After euthanasia, a full post-mortem examination was conducted and several tissue samples, including tonsils, were collected and frozen at −70 °C until tested. A portion of tissues were collected in formalin buffered saline for histopathological analysis. 

### 2.7. Hematological and Biochemical Analysis

Freshly collected whole blood in sodium citrate and serum samples was subjected to hematological and biochemical analysis, respectively. Hematological analysis was performed on a Vetscan^®^ HM5 analyzer (Abaxis, Inc., Union City, CA, USA) with the following parameters evaluated: red blood cells, hemoglobin, hematocrit, mean corpuscular volume, mean corpuscular hemoglobin, mean corpuscular hemoglobin concentration, red cell distribution weight, platelets, mean platelet volume, white blood cells, neutrophil count [absolute (abs) and %], lymphocyte count (abs and %), monocyte count (abs and %), eosinophil count (abs and %), and basophil count (abs and %). Serum chemistries were evaluated on the Vetscan^®^ HM5 Hematology Analyzer using VetScan^®^ Comprehensive Diagnostic Profile reagent rotors (Abaxis, Inc.) with the following parameters evaluated: alanine aminotransferase, albumin, alkaline phosphatase, amylase, calcium, creatinine, globulin, glucose, phosphorus, potassium, sodium, total bilirubin, total protein, and blood urea nitrogen. 

### 2.8. Antibody Detection

An enzyme-linked immunosorbent assay (ELISA) Test Kit IDEXX CSFV Ab Test (Catalogue number 99-43220, IDEXX Laboratories, Markham, ON, Canada) was utilized to detect anti-CSFV antibodies in serum samples. All samples were tested in duplicate, following the manufacturer’s overnight incubation protocol. The sample absorbance was measured at 450 nm using a SpectraMax^®^ microplate reader (Molecular Devices, LLC., San Jose, CA, USA) and the SoftMax Pro software version 7.0 (Molecular Devices). Blocking percentages were calculated according to the manufacturer’s protocol. Samples with blocking percentages ≥40% were considered positive for antibodies to CSFV, and the samples with blocking percentages of ≤30% were considered negative. Samples in between the two cut-offs were considered suspect and re-tested. 

Samples that were positive or suspicious on ELISA were tested by the NCFAD Neutralization Peroxidase-Linked Assay (NPLA) assay. The challenge virus CSFV Alfort/187 was used. Briefly, the test serum was added to a 96-well sterile Nunc plate at a 1/5 dilution in duplicate. The challenge virus was then added, and a two-fold serial dilution was performed down the plate and incubated for one hour at 5% CO_2_ at 37 °C. Afterwards, the plates were removed and PK-15 cells in growth media supplemented with 20% irradiated horse serum (Thermo Fisher Scientific) were added. The plates were incubated for 3 days at 5% CO_2_ at 37 °C, fixed, and stained using anti-CSFV polyclonal pig serum and HRP-conjugated goat-anti-pig polyclonal serum. The absence of the dark red positive staining would indicate a lack of CSFV antigen expression; therefore indicating the presence of CSFV-specific neutralizing antibodies in the serum samples tested.

### 2.9. Histopathology and Immunohistochemistry (IHC)

Tissues fixed in 10% neutral phosphate buffered formalin were sectioned at 5 µm and stained with hematoxylin and eosin (H &E) for histopathologic examination. For IHC, paraffin tissue sections were quenched for 10 min in aqueous 3% hydrogen peroxide, and the epitopes were retrieved using Glyca Target Retrieval solution (made in-house) in a Decloaking Chamber™ (Biocare Medical, Pacheco, CA, USA). The sections were treated with the monoclonal anti-CSFV antibody WH303 [48,49,50] at a 1:500 dilution for 30 min. The sections were then visualized using EnVision+ Single Reagent HRP Mouse kit (Agilent Technologies Canada Inc., Mississauga, ON, Canada) and reacted with the chromogen diaminobenzidine + (DAB+) substrate (Agilent Technologies Canada Inc.). Finally, the sections were counter-stained with Gill’s hematoxylin (made in-house). 

## 3. Results

### 3.1. Laboratory Analysis of the Clinical Samples from Colombia

Of the twenty-three samples received from Colombia, seventeen samples tested positive by RRT- PCR (Table 1). Twelve out of the seventeen samples yielded virus isolates. Seven isolates were selected for whole genome sequencing. All seven samples selected resulted in complete genome sequences and, during BLASTN search, closely matched with the CSFV strain Nghe An/2013 from Vietnam (GenBank: LC388757), which was classified as a genotype 2.2 strain at the time of the submission (14 June 2018). However, based on the recently proposed nomenclature, CSFV Nghe An/2013 and the Colombian isolates now belong to the genotype 2.6 (Figure 2). 

The genotype 2.6 node also included CSFV strains isolated from China in 2017 (MT586092), Vietnam in 2010 (MZ643989), 2014 (KP702207, KP702210, MH979232), and 2017 (MZ869044, MZ869046), Indonesia in 2007 (EU180068), Thailand in 2001 (MK026463, MK026462), and Italy in 1998 (JQ411579). The whole genome sequences of the seven Colombian CSFV isolates from 2013, 2014, 2016, and 2017 were highly similar to each other (≥98.6%) and formed a well-supported monophyletic clade, indicating that it is the same virus circulating in various Colombian departments. The 190 nucleotide stretch of the E2 hypervariable region of these isolates was also highly similar (≥98.42%) to the previously reported CSFV genotype 2 isolates from Colombia in 2005 and 2006 (GenBank: DQ338466 and EF565743, respectively) [31]. Our mugration model inference of ancestral countries of origin across the phylogeny suggest that the Colombian CSFV isolates are derived from Vietnam with >0.99 support. 

### 3.2. Experimental Inoculation—Clinical Signs and Clinical Scores

All five pigs (#16–20) infected with CSFV 2016/Pinillos developed fever. The first pig to develop fever (defined as a rectal temperature above 40 °C for two consecutive days) was pig #16, 4 days post-infection (dpi), followed by the rest of the pigs on 5 dpi (Figure 3). The rectal temperatures fluctuated but the fever lasted in all pigs until the end of the study. Starting at 6 dpi, some dry feces were observed in the pen, and the amount increased as the disease progressed. On 9 dpi, pigs were less active and small cutaneous hemorrhages started to appear on pigs #17, 18, and 20. Approximately ¼ of the feed was left in the bins, and pig #20 was slow to recover from anesthesia. On 10 dpi, there was less feces in the pen than normal, urine was highly concentrated, and pig #20 developed a cough. All pigs were slow to rise and returned to laying down very quickly. On 15 dpi, pig #17 was ataxic. On 16 dpi, pigs #17 and #18 had cutaneous hemorrhages in the ears and snout, and showed slow recovery from anesthesia. By 17 dpi, pigs were playful and regained appetite, as there was no feed left from the prior day. On 19 dpi, pig #16 was ataxic. On 20 dpi, pigs were depressed again, and pig #16 developed widespread cutaneous hemorrhages and the rectal temperature dropped to 36 °C (hypothermia), and later in the day it was euthanized. As pigs were not interested in feed, they were offered fresh fruit and veggies. On 21 dpi, pig #20 also developed extensive cutaneous hemorrhages, rectal bleeding, and labored breathing, and therefore it was euthanized. On 22 dpi, all three remaining pigs (#17, 18 and 19) appeared pale. Pigs #17 and 18 developed extensive cutaneous hemorrhages, and pig #18 had piloerection and was shivering. Later in the day, these three pigs were euthanized. No diarrhea was observed throughout the study period. 

The clinical score, which indicates the seriousness of the illness, was calculated for each animal every day, as seen in Figure 3. In these animals, the clinical scores started to rise around 5 dpi and peaked around 12–13 dpi. The scores improved around 16 dpi, and started to increase again starting at 17 dpi. Until the end of the experiment, the average daily clinical score for all the animals remained below 14. 

### 3.3. Experimental Inoculation—CSFV Genome Detection

CSFV genomic material was first detected on 2 dpi in whole blood collected from pig #16 with a Ct value of 39 (Figure 4A). By 5 dpi, whole blood samples from all five pigs tested positive for CSFV genomic material, with Ct values ranging from 29 to 31. Viremia peaked around 12 dpi in all animals and remained high until the end of the study. CSFV genomic material was detected in nasal and oropharyngeal (OP) swabs from all animals, starting at 5 dpi (Figure 4B). Ct values ranged from 32 to 36 in oropharyngeal swabs, and from 37 to 39 in nasal swabs. In contrast, only 2/5 buccal swabs tested positive on 5 dpi (pig #16 and pig #17). By 9 dpi, all swab types from all five of the pigs tested positive. The lowest Ct values observed (highest viral load) for each swab type were the following: 20.35 (buccal, pig #20, dpi 21), 18.16 (oropharyngeal, pig #20, dpi 21) and 19.21 (nasal, pig #18, dpi 21).

At the end of the experiment, different sample types were collected during the post-mortem examination. All tissue samples tested positive for CSFV genomic material (Figure 4C), but skin, ear notches, and brain tissues had lower levels of CSFV genomic material compared to those in the lymphoid organs. 

### 3.4. Experimental Inoculation—Serology

The presence of antibodies to CSFV in serum samples was measured using a blocking ELISA. All pigs except one (pig #20) stayed sero-negative throughout the experiment (Figure 5). The last two serum samples collected from pig #16 showed suspected levels of antibodies to CSFV. The positive and two suspected samples were subjected to confirmatory NPLA, but no neutralizing antibodies to CSFV were detected in those samples.

### 3.5. Experimental Inoculation—Hematology

Severe thrombocytopenia was observed in three out of five piglets as early as 2 dpi and continued to reduce. Leukopenia with lymphopenia and neutropenia was also noticed (Figure 6). From about 7 dpi onwards, anemia, was apparent with a drastic reduction in red blood cell counts, hemoglobin levels, and hematocrit. In terms of clinical chemistry, we did not observe any significant findings, except serum albumin levels that decreased slowly with time. 

### 3.6. Experimental Inoculation—Gross Pathology

All five pigs were subjected to full post-mortem examinations. Pigs #16, #18, and #20 had extensive cutaneous hemorrhages (petechial and ecchymotic) on the ears, neck, abdomen, and legs (Figure 7A–D). The lungs of pig #17 showed severe consolidation of the caudal lobes with very little normal tissue remaining (Figure 7J). Similar, but less severe lung lesions were also seen in pigs #16 and #20. All pigs had increased amounts of pericardial fluids, and hemorrhagic pericarditis. The majority of the lymph nodes (Figure 7E,P) were enlarged and hemorrhagic. The kidneys showed medullary hemorrhages (Figure 7O), and petechial cortical hemorrhages (Figure 7K). The blood of pigs #16 and #18 was notably thin and hemorrhages were seen in the skeletal muscle of pig #20 (not shown). Also noted were hemorrhages in the epiglottis (Figure 7F), heart (Figure 7I), gastric mucosa (Figure 7M), cecum (Figure 7N), and ileocecal valves (Figure 7L) of some pigs. 

### 3.7. Experimental Inoculation—Histopathology and Immunohistochemistry

In the lymph nodes, extensive hemorrhages in the medulla were observed (Figure 8A,B). CSFV E2 antigen was detected primarily within the dendritic cells in the germinal centers (Figure 8C,D). In the lungs, the alveolar walls were mildly expanded by the presence of inflammatory cells (Figure 8E) and several fibrin micro-thrombi were observed (Figure 8F). Positive immunostaining for CSFV E2 was primarily observed within the bronchiolar epithelium (Figure 8G,H). In the spleen, there were prominent peri-arteriolar lymphatic sheaths, and numerous cells with pyknotic nuclei indicating necrosis in adjacent red pulp. There was a decrease in white pulp and a lack of germinal centers. 

CSFV E2 positive staining was primarily noticed in peri-arteriolar lymphatic sheaths. In the ear notches and skin biopsies, CSFV E2 protein was detected in epidermal epithelial cells and hair follicles (Figure 9A,B). There were no lesions in epidermal epithelial cells and hair follicles observed using H&E staining. In the tonsillar crypts, necrotic crypt epithelial cells (Figure 9C) were observed and the cells were positively stained for CSFV E2 (Figure 9D). In the bone marrow, a large number of pyknotic nuclei were observed (Figure 9E) and the bone marrow cells were strongly positive for CSFV E2 antigen (Figure 9F). In the kidneys, multifocal hemorrhages were observed in the medulla with a degeneration of tubular epithelial cells in adjacent areas (Figure 9G,H). CSFV E2 antigens were detected in renal tubular epithelial cells and epithelial cells of the renal pelvis (I and J). At the ileocecal junction, submucosal hemorrhages and edema were observed and CSFV E2 antigen positive cells were observed in germinal centers. In the urinary bladder, submucosal hemorrhages and mild erosions were observed, and the basal epithelial layer consistently stained for CSFV E2 antigen. In the heart, multifocal hemorrhages were observed but no CSFV E2 antigens were detected. In the stomach, multifocal mucosal, submucosal, and serosal hemorrhages were observed and tubular epithelial cells were positively stained for CSFV E2 antigen. In the colon, multifocal hemorrhages in the mucosa and submucosa were observed but no staining for CSFV E2 antigens. Most brain sections had no lesions, but a few instances of perivascular cuffing by inflammatory cells was observed. No staining for CSFV E2 was observed. 

## 4. Discussion

According to the national swine census published by the Instituto Colombiano Agropecuario “ICA”, Colombia housed around 6.7 million pigs in 2019, in approximately 240,000 swine production centres. One third of those farms were backyard systems and only two thirds were ‘technified’ farms. More than half of the pigs in the country are housed in Antioquia, Cundinamarca, and Valle del Cauca [51]. CSF has been endemic in Colombia since 1942 [29]. In 2000, Colombia initiated a National Program for the Eradication of CSF based on the extensive vaccination, monitoring, and elimination of infected herds [52]. The program drastically reduced the CSF outbreaks, and the 17 departments in Colombia have been classified into three zones: the WOAH-recognized CSF free zone (central-west and the central-east regions), the self-declared CSF free zone (south-east region), and the CSF control zone in the north [53,54]. Despite continuous mass vaccination in the CSF control zone, two major CSF outbreaks have been reported within the past two decades. During the first outbreak, which occurred between 2005 and 2006, CSFV genotype 2 was identified from domestic pigs in the La Guajira and Norte de Santander departments [31]. Since then, there had been no reports of presence of this genotype in the region [31]. 

Our results confirm that the second CSF outbreak that occurred between 2013 and 2018 in northern Colombia was also caused by a CSFV genotype 2 strain. The virus responsible for 2005–2006 was not fully sequenced; however, based on the 190 bp sequence of the E2 hypervariable region available for two CSFV isolates from 2005 to 2006 [31], it is highly likely that both outbreaks were caused by a virus originating from a common source. Colombia and Venezuela share a common border and the illegal movement of pigs and pork products is common due to the difference in pig prices between the two countries. Therefore, the illegal introduction of CSFV-genotype-2-infected pigs or pig products from Venezuela could be responsible for the two outbreaks. The CSF status in Venezuela is unknown. 

When the newly proposed nomenclature is used, based on the full-length E2 gene sequences, 2013–2018 CSFV Colombia isolates can be grouped into genotype 2.6, together with CSFV strains isolated from Vietnam in 2010, 2013, 2014, and 2017, Thailand in 2001, Indonesia in 2007, China in 2017, and Italy in 1998. As observed with the CSFV 2016/Pinillos strain, the genotype 2.6 Vietnam strains also induced acute infection in 4-week-old piglets, and were classified as moderately virulent strains [55,56]. 

When inoculated into 6-week-old weaned piglets, the CSFV 2016/Pinillos strain induced fever in all pigs by 5 dpi, displaying an acute course of infection as described with other moderately virulent strains. Pigs infected with CSFV 2016/Pinillos lived up to 21–22 dpi until they were euthanized. The fever remained above 40 °C in all animals, with some pigs reaching 41 °C. Despite the fever, pigs remained alert and active until 9 dpi. Some pigs developed constipation, but no diarrhea was observed. Cutaneous hemorrhages were observed starting around 16 dpi onwards in all pigs and they were euthanized on 21 and 22 dpi as they reached humane end points. Based on the daily clinical scores, the CSFV 2016/Pinillos strain was labeled a moderately virulent strain. In contrast, highly virulent CSFV strains have killed pigs in this age group within 7–10 dpi [6]. 

As described for acute CSFV infections, all pigs infected with CSFV 2016/Pinillos showed decreasing numbers of lymphocytes, granulocytes, and platelets as early as 2 dpi [57,58,59,60,61,62]. Total WBC counts, neutrophils, lymphocytes, and platelets (4/5 pigs) dropped below normal levels by 5 dpi and remained low until the end of the study; the majority of the pigs became anemic by 9 dpi. 

The CSFV 2016/Pinillos strain induced gross and histopathological lesions similar to those reported before for other moderately virulent CSFV strains, including the genotype 2.6 strains from Vietnam [55,63]. 

The first case of CSFV genotype 2.6 (historically classified as genotype 2.2) in South-East Asia was reported in 2001 from Thailand, and later from the neighboring countries [64,65,66]. The appearance and efficient spread of CSFV genotype 2.6 in Asia, likely from an ancestral CSFV strain that circulated in Italy in 1998, indicates the continuous evolution of CSFV in the region, reducing the effectiveness of the current vaccination efforts [67]. No reports of CSFV genotype 2 were present in South America before 2005. Therefore, it is highly likely that CSFV genotype 2.6 was introduced to South America from South-East Asia, which is supported by our discrete character mugration inference. The presence of a moderately virulent CSFV genotype 2.6 strain in Colombia, and most likely in Venezuela, complicates the CSFV eradication efforts in the Americas. As a part of the FAO-led CSF eradication effort, many countries conduct mass vaccination campaigns using attenuated C-strain (genotype 1) vaccine. It is well known that the CSFV vaccines induce neutralizing antibodies to homologous strains better than the heterologous strains. Partial protection induced by the ongoing vaccination might alter the current clinical signs leading to asymptomatic or chronic cases. Therefore, ongoing vaccination efforts may not be able to effectively control the spread of genotype 2.6 virus and promote viral evolution. 

## 5. Conclusions

In summary, the northern departments of Colombia experienced two major CSF outbreaks caused by a recently introduced moderately virulent genotype 2.6 CSFV. The phylogenetic analysis of viral sequences suggests a possible recent transboundary spread of the virus from South-East Asia to South America. This study emphasizes the need for both active and passive surveillance, to monitor the spread of this new CSFV genotype in Colombia and to the neighboring countries, in order to successfully eradicate CSF from the Americas.

## Figures and Tables

**Figure 1 viruses-15-02308-f001:**
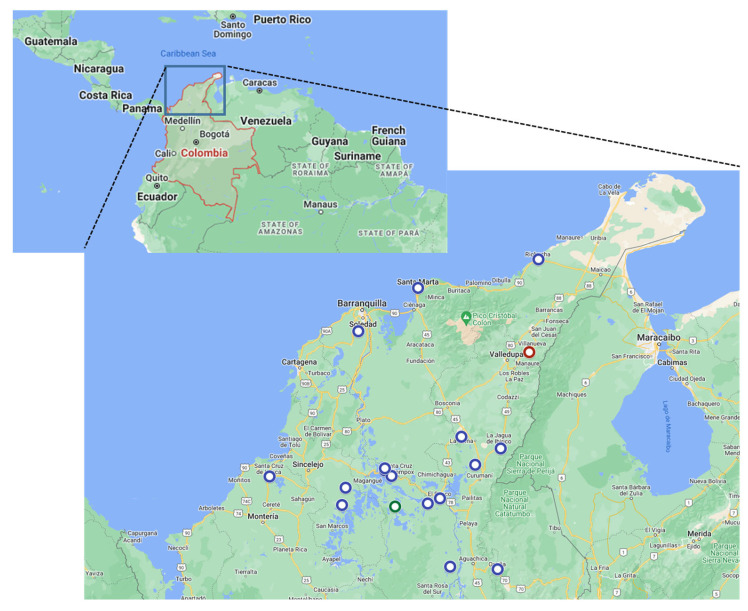
Map of Colombia showing the geographic locations (circles) of samples used in this study. Red circle: location of the index farm. Green circle: location of the CSFV Pinillos/2016 isolate. Figure generated using Google My Maps—https://mymaps.google.com/ (accessed on 1 September 2023).

**Figure 2 viruses-15-02308-f002:**
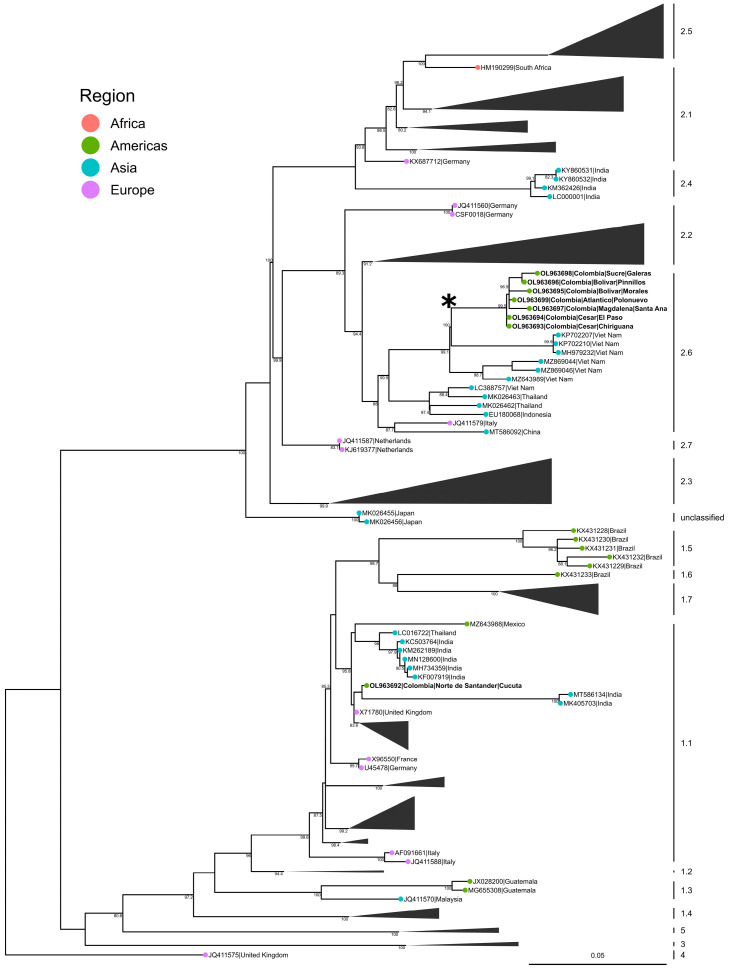
Phylogenetic analysis of available full-length CSFV E2 sequences. Bold: sequences from this study and a historic CSFV isolate from Colombia. Some of the large nodes were collapsed for easy visualization of the Colombian strains with closely related sub-genotypes. The CSFV Congenital Tremor was used as the outgroup (JQ411575). Bootstrap (SH-aLRT) values greater or equal than 80 are shown. Asterisk indicates discrete character reconstruction of “Viet Nam” at the adjacent node. For the original, uncollapsed tree, please see Appendix A.

**Figure 3 viruses-15-02308-f003:**
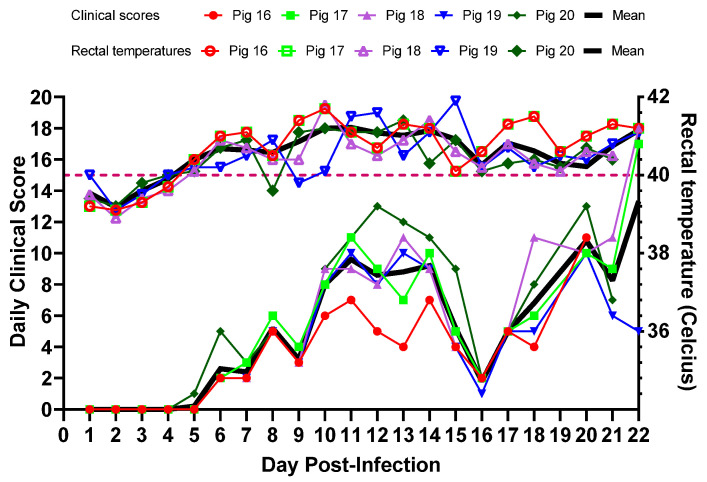
The daily clinical scores (closed shapes) and the daily rectal temperatures (open shapes) of pigs infected with the CSFV 2016/Pinillos strain.

**Figure 4 viruses-15-02308-f004:**
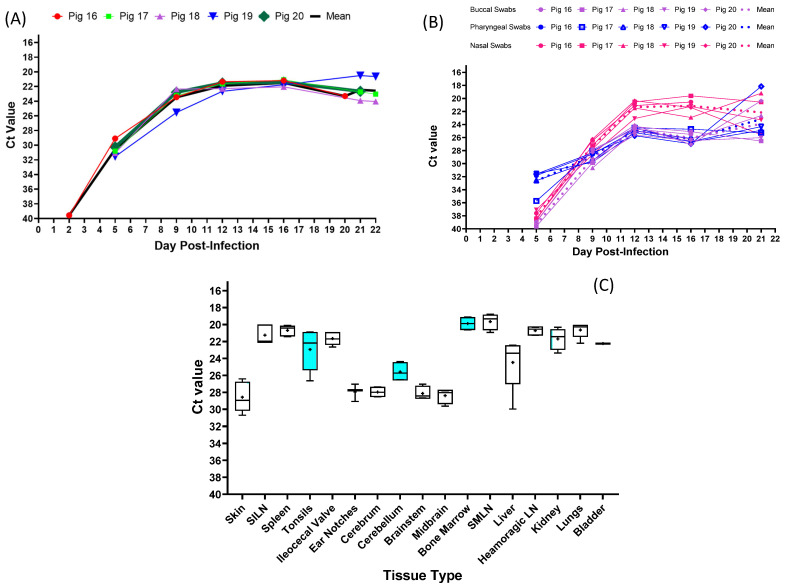
Detection of CSFV genomic material in clinical samples collected from pigs inoculated with the CSFV 2016/Pinillos: (**A**) whole blood; (**B**) swabs; (**C**) tissues.

**Figure 5 viruses-15-02308-f005:**
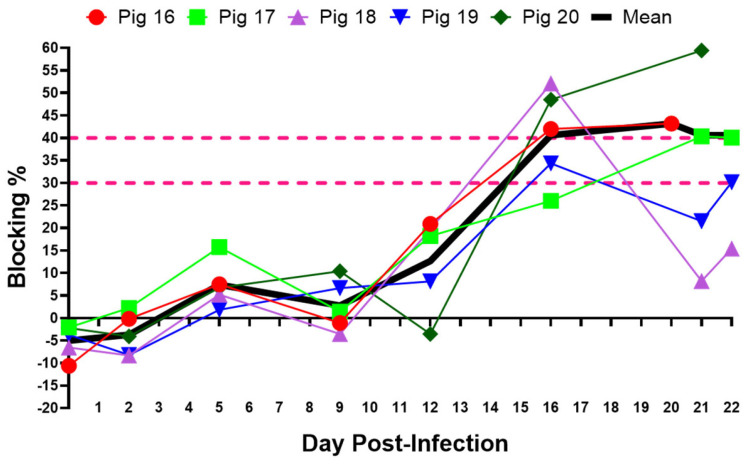
Antibody response in pigs infected with the CSFV 2016/Pinillos. The cut-off values for the ELISA are indicated by dotted lines (<30%, negative; 30–40%, doubtful; ≥40%, positive).

**Figure 6 viruses-15-02308-f006:**
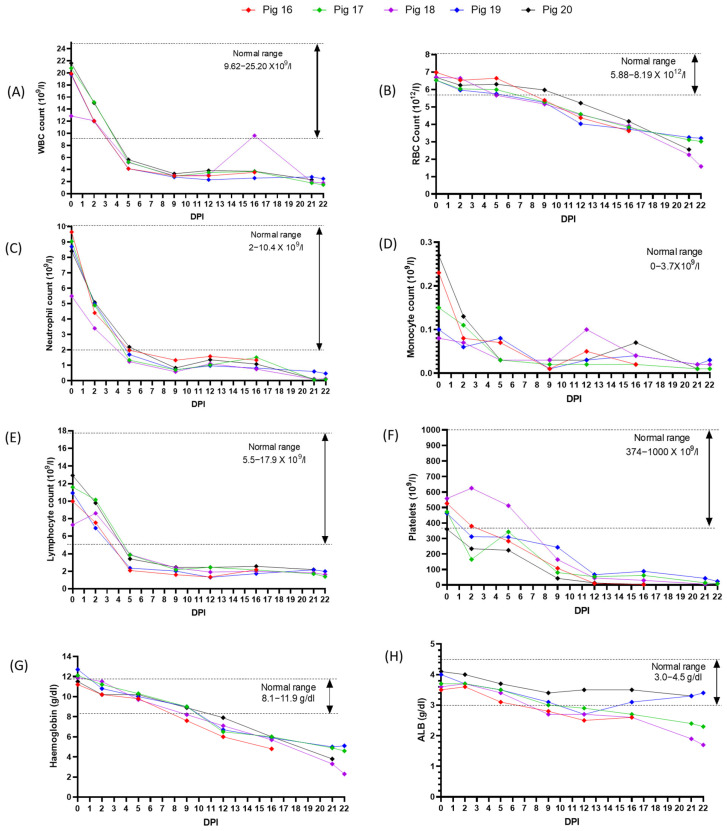
Hematological parameters of pigs infected with CSFV Pinillos/2016. Various elements captured included (**A**) white blood cell count; (**B**) red blood cell count; (**C**) neutrophil count; (**D**) monocyte count; (**E**) lymphocyte count; (**F**) platelet count; (**G**) hemoglobin count; (**H**) albumin.

**Figure 7 viruses-15-02308-f007:**
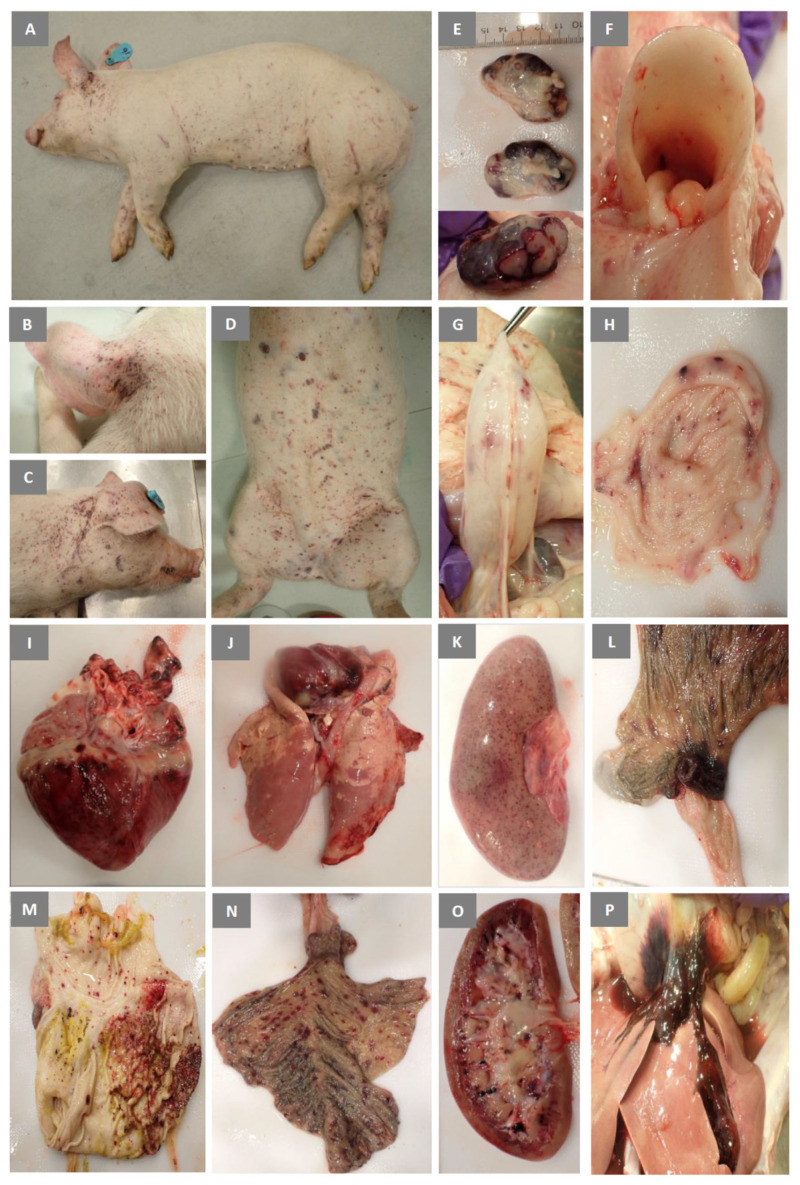
Gross pathology observed in pigs inoculated with CSFV Pinillos/2016. Petechial and ecchymotic hemorrhages in the skin (**A**–**D**). Enlarged submandibular lymph nodes with multifocal hemorrhages (**E**). Severe pneumonia involving the caudal lobes (**J**). Hemorrhages in the (**F**) epiglottis, serosal (**G**), and mucosal (**H**) surface of the urinary bladder, (**I**) epicardium, (**K**) renal cortex, (**L**) ileocecal junction, (**M**) gastric mucosa, (**N**) cecal mucosa, (**O**) renal medulla, and (**P**) gastro-hepatic lymph node and gall bladder.

**Figure 8 viruses-15-02308-f008:**
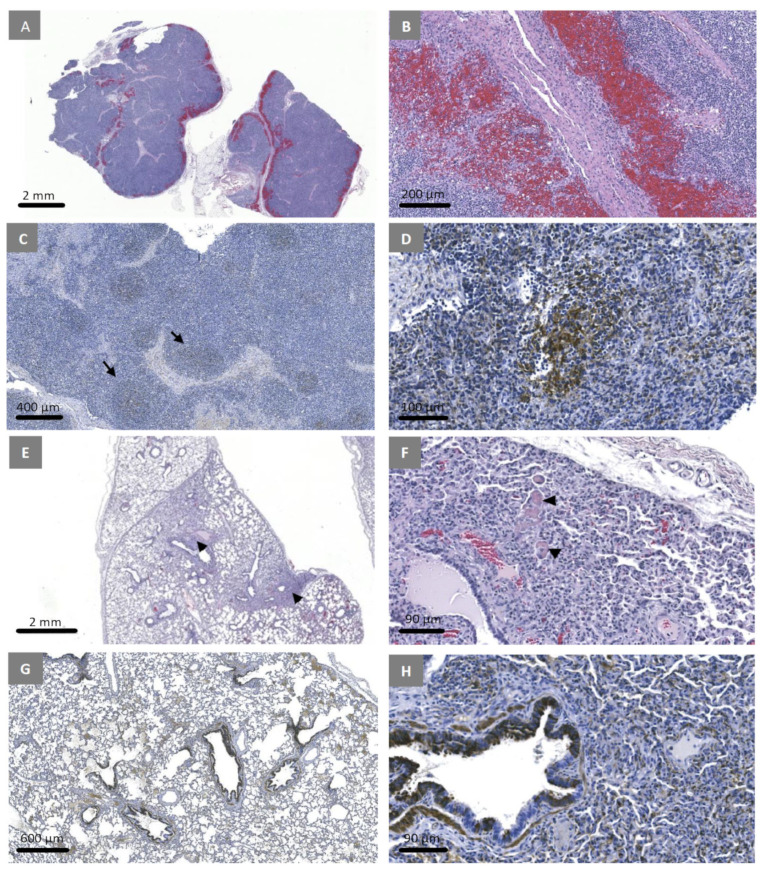
Histopathology and immunohistochemical findings in pigs inoculated with CSFV Pinillos/2016. Extensive hemorrhage in the medulla of submandibular lymph nodes (**A**,**B** [higher magnification]). Viral antigens detected within germinal centres of lymphoid follicles (**C** [arrows], **D** [higher magnification]). Multifocal interstitial pneumonia (**E**, arrowheads). Multiple fibrin microthrombi in lung capillaries (**F**, arrow heads). Viral antigens primarily detected in bronchiolar epithelial cells (**G**,**H** [higher magnification]). (**A**,**B**,**E**,**F**—H&E stain, **C**,**D**,**G**,**H**—immunohistochemistry).

**Figure 9 viruses-15-02308-f009:**
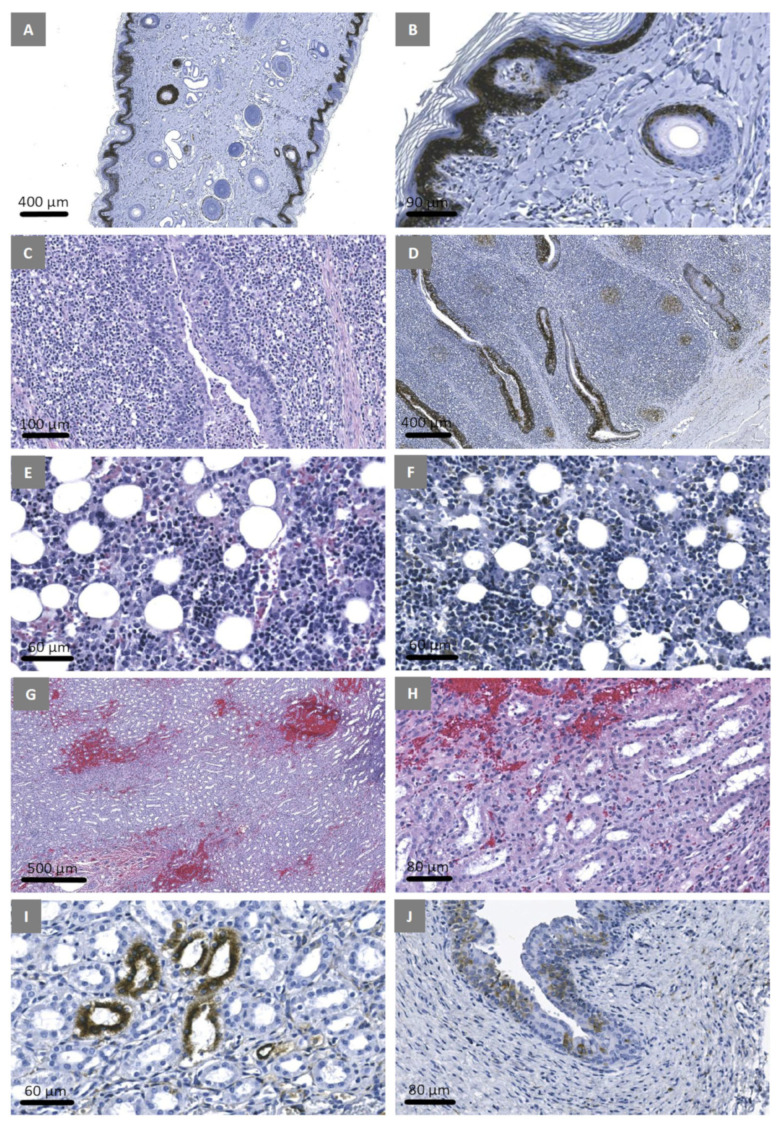
Histopathology and immunohistochemical findings in pigs inoculated with CSFV Pinillos/2016. CSFV E2 antigen in the dermis and hair follicles in the ear notches (**A**,**B** [higher magnification]). Necrosis of epithelial cells in tonsillar crypts (**E**) associated with positive immunostaining for CSFV E2 (**D**). Renal medulla with multifocal hemorrhages (**G**) and adjacent degeneration of tubular epithelial cells (**H**). CSFV E2 antigens in renal tubular epithelial cell (**I**) and epithelial cells of the renal pelvis (**J**). (**C**,**E**,**G**,**H**—H&E stain, **A**,**B**,**D**,**F**,**I**,**J**—immunohistochemistry).

**Table 1 viruses-15-02308-t001:** History and summary of laboratory results for the samples tested.

Sample	Sample #	Type	Department	Municipality	Type of Operation	Date of Collection	RRT-PCR	GenBank #
1	2013-409S	Serum	La-Guajira	Urumita	Backyard	1 June 2013	0.00	
2	2013-409T	Tonsil	La-Guajira	Urumita	Backyard	1 June 2013	33.36	
4	2013-1070	Serum	Cesar	El-Paso	Backyard	3 September 2013	35.65	
3	2013-922	Lymph Node	Cesar	Chiriguana	Backyard	29 September 2013	0.00	
5	2013-1185	Serum	Cesar	Chiriguana	Backyard	2 November 2013	18.75	OL963693
6	2014-58	Serum	Cesar	El Paso	Backyard	24 January 2014	26.15	OL963694
7	2014-257	Spleen	Magdalena	Santa-Ana	Backyard	7 March 2014	25.05	
8	2014-519	Tonsil	Cesar	Chiriguana	Backyard	11 April 2014	24.69	
9	2014-598	Kidney	Boliver	Talaigua-Nuevo	Backyard	12 May 2014	26.85	
10	2014-947	Serum	Norte-Santander	Ocana	Backyard	9 August 2014	0.00	
11	2014-1343	Serum	Magdalena	San-Zenzon	Backyard	11 October 2014	26.49	
12	2015-113	Serum	Magdalena	El-Banco	Backyard	1 February 2015	0.00	
13	2015-246	Tonsil	La Guajira	Riohacha	Backyard	1 March 2015	36.90	
14	2015-389	Serum	Sucre	San-Benito-Abad	Backyard	5 March 2015	28.12	
15	2015-302	Serum	Cesar	La-Jagua-De-Ibirico	Backyard	10 March 2015	0.00	
16	2015-338	Tonsil	Magdalena	Santa-Marta	Backyard	21 March 2015	34.09	
17	2015-915	Serum	Bolivar	Barranco -De-Loba	Backyard	10 July 2015	0.00	
18	2016-11	Tonsil	Bolivar	Morales	Backyard	2 January 2016	22.59	OL963695
19	2016-666	Serum	Cordoba	Lorica	Backyard	8 June 2016	25.26	
20 *	2016-686	Serum	Bolivar	Pinnillos	Backyard	4 July 2016	22.36	OL963696
21	2016-860	Serum	Magdalena	Santa Ana	Backyard	24 July 2016	23.52	OL963697
22	2017-356	Lymph Node	Atlantico	Polonuevo	Commercial	28 April 2017	23.74	OL963699
23	2017-848	Lymph Node	Sucre	Galeras	Backyard	15 September 2017	22.52	OL963698

* The sample from which the CSFV Pinillos/2016 was isolated.

## Data Availability

The authors will provide the data related to this manuscript upon request.

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
