# Peer review of "Molecular and Pathological Characterization of Classical Swine Fever Virus Genotype 2 Strains Responsible for the 2013–2018 Outbreak in Colombia"

_viruses, 2023, doi:10.3390/v15122308_

Round 1

Reviewer 1 Report

Comments and Suggestions for Authors

Dear Editor,

Thank you for the opportunity to review the manuscript submitted to the "Viruses" journal by Robert et al. The work aligns well with the journal's scope, offering a descriptive analysis of a notably devastating virus affecting animal health. The authors adeptly characterize the outbreaks in Colombia using a variety of tools, such as sequencing, phylogenetic analysis, and both in vitro and ex-vivo methods.

While the manuscript is commendably written and certainly merits publication, I would like to highlight two significant concerns that, in my professional judgment, have not been adequately addressed:

Major Issues:

The authors have mistakenly linked the strains circulating in Colombia with those from Asia. This is a critical point. To my knowledge, no reliable study has demonstrated a correlation based on genetic distance or ancestral lineage with geographical structure in classical swine fever. To substantiate such a claim, the authors should employ a model reconciling the discrete states (geographical location) with the internal nodes and branch lengths of their phylogenetic trees. To my knowledge, the two suitable models are the mugration model from TreeTime, grounded in Maximum Likelihood, and the discrete model from BEAST1.10, which operates on Bayesian inference. Without such analyses, the authors are essentially presenting a Maximum Likelihood tree with mapped location samples for the terminal tip, making no association between location (internal nodes) and dispersal (tree tips). Given that this aspect is a central conclusion in the manuscript, without proper analytical backing, I recommend that these claims either be substantiated with rigorous analysis or removed entirely to avoid potential scientific criticism and the risk of manuscript retraction.

My second major concern pertains to the phylogenetic analysis itself. Upon examining Figure 2, I was surprised to find the tree lacking statistical support for the bifurcation between subgenotypes 1.3 and other subgenotypes of Genotype 1 of CSFV. GI suspect this issue arises from the choice of strains used in the topological reconstruction. I urge the authors to clarify this point. Additionally, for the sake of transparency and reader comprehension, the complete tree with all taxa should be provided in the supplementary information. The authors could then clarify that certain clades were collapsed for visual clarity. The omission of some subgenotypes in the tree presented in Figure 2 is potentially misleading and should be addressed.

Minor Comment:

The authors state that they compared the 190-E2 sequences with whole-genome sequences but did not specify the method used to obtain these sequences. Was it through NGS with low coverage, or did they utilize Sanger sequencing? Why were different methods chosen?

I trust that my comments will assist the authors in refining their manuscript. I firmly believe the study warrants publication once the aforementioned issues are rectified.

Author Response

Reviewer 1

Thank you for the opportunity to review the manuscript submitted to the "Viruses" journal by Robert et al. The work aligns well with the journal's scope, offering a descriptive analysis of a notably devastating virus affecting animal health. The authors adeptly characterize the outbreaks in Colombia using a variety of tools, such as sequencing, phylogenetic analysis, and both in vitro and ex-vivo methods.

 While the manuscript is commendably written and certainly merits publication, I would like to highlight two significant concerns that, in my professional judgment, have not been adequately addressed:

We thank the reviewer for the kind feedback about our work.

 Major Issues:

The authors have mistakenly linked the strains circulating in Colombia with those from Asia. This is a critical point. To my knowledge, no reliable study has demonstrated a correlation based on genetic distance or ancestral lineage with geographical structure in classical swine fever. To substantiate such a claim, the authors should employ a model reconciling the discrete states (geographical location) with the internal nodes and branch lengths of their phylogenetic trees. To my knowledge, the two suitable models are the mugration model from TreeTime, grounded in Maximum Likelihood, and the discrete model from BEAST1.10, which operates on Bayesian inference. Without such analyses, the authors are essentially presenting a Maximum Likelihood tree with mapped location samples for the terminal tip, making no association between location (internal nodes) and dispersal (tree tips). Given that this aspect is a central conclusion in the manuscript, without proper analytical backing, I recommend that these claims either be substantiated with rigorous analysis or removed entirely to avoid potential scientific criticism and the risk of manuscript retraction.

We thank the reviewer for pointing this out. As the reviewer suggested we reanalyzed the sequences using  the mugration model from TreeTime (v0.11.1) The analysis supports the relationship between the Colombian isolates with the  CSFV strains from Vietnam (2014). Therefore we did not remove our statements related to the possible spread of CSFV Genotype 2.6 strain from Asia to Colombia.   

My second major concern pertains to the phylogenetic analysis itself. Upon examining Figure 2, I was surprised to find the tree lacking statistical support for the bifurcation between subgenotypes 1.3 and other subgenotypes of Genotype 1 of CSFV. GI suspect this issue arises from the choice of strains used in the topological reconstruction. I urge the authors to clarify this point. Additionally, for the sake of transparency and reader comprehension, the complete tree with all taxa should be provided in the supplementary information. The authors could then clarify that certain clades were collapsed for visual clarity. The omission of some subgenotypes in the tree presented in Figure 2 is potentially misleading and should be addressed.

We thank the reviewer for this valuable suggestion. In order to clarify this issue, we repeated the  phylogenetic analysis using CSFV Congenital Tremor UK (JQ411575) as the outgroup, and replaced Figure 2. We also included a complete tree with all taxa with no collapsed clades as a supplementary figure (Figure S1).

Minor Comment:

The authors state that they compared the 190-E2 sequences with whole-genome sequences but did not specify the method used to obtain these sequences. Was it through NGS with low coverage, or did they utilize Sanger sequencing? Why were different methods chosen?

The 190 E2 sequence information related to the Colombia in 2005 and 2006 outbreak was obtained from the GenBank. According to the publication, they were obtained using Sanger method (Garrido Haro AD, Barrera Valle M, Acosta A, J Flores F. Phylodynamics of classical swine fever virus with emphasis on Ecuadorian strains. Transbound Emerg Dis. 2018 Jun;65(3):782-790). All the sequence information related to the  2013-2018 outbreak were obtained by us, using the Ion S5 system (whole genome sequencing).

I trust that my comments will assist the authors in refining their manuscript. I firmly believe the study warrants publication once the aforementioned issues are rectified.

We appreciate the valuable and constructive comments from the reviewer. We strongly believe that the suggested revisions improved this manuscript.

Reviewer 2 Report

Comments and Suggestions for Authors

This paper contains valuable and rare insights into the pathological changes caused by CSFV. 

In order to provide a more focused and relevant description related to research findings, it is recommended to omit unnecessary details and maintain a concise format. The inclusion of pertinent information regarding the materials and methods will enhance the clarity and effectiveness of your write-up, especially the introduction and discussion parts.

In the materials and methods, please mention the company, city, and country when the material was mentioned.

There are too many references. Consider to reduce the number into the most relevant and impactful references that directly support your research.

Author Response

Comments and Suggestions for Authors

This paper contains valuable and rare insights into the pathological changes caused by CSFV. 

We thank the reviewer for supporting our manuscript.

In order to provide a more focused and relevant description related to research findings, it is recommended to omit unnecessary details and maintain a concise format. The inclusion of pertinent information regarding the materials and methods will enhance the clarity and effectiveness of your write-up, especially the introduction and discussion parts. In the materials and methods, please mention the company, city, and country when the material was mentioned.

We have included this information. 

There are too many references. Consider to reduce the number into the most relevant and impactful references that directly support your research.

We have reduced the number of references (84 to 73).

Reviewer 3 Report

Comments and Suggestions for Authors

Major comments

Material and methods

-        You should report the approval number of this study by an ethical committee of your institute or university.

-        L107-108: Did you collect the samples from live or dead animals? Give more details about the animals (e.g., age, breed, etc.).

-        L157-170: Add appropriate references.

-        L213-220: Add a table with the used scoring system, giving details per score 0, 1, 2, and 3 and adding the appropriate reference.

Discussion 

-        You could include a paragraph of thoughts on the economic impact of your results on the swine industry in South Asia.

Minor comments

-        L21:.. the full-length..

-        L36: .. Paraguay, Uruguay, and some areas

-        L52: .. into high, moderate, and low..

-        L79: .. because of trans-placental transmission

-        L129: … in each homogenizing..

-        L130: .. 10-second runs with 10-second pauses

-        L135: … homogenate were used for..

-        L260: .. during the lactation phase….

-        L159: .. CSFV-infected PK-15 cell…

-        L212: .. clinical findings of each animal..

-        L223: .. On each sampling day…

-        L581: One-third of those farms..

-        L582: More than half of the pigs..

-        L609: .. 6-week-old weaned piglets..

-        L612: The fever remained above..

-        L613: Despite the fever..

-        L633: .. the FAO-led CSF eradication..

-        L641: .. the Northern departments..

-        L643: .. sequences suggests possible..

Author Response

Major comments

Material and methods

-        You should report the approval number of this study by an ethical committee of your institute or university.

This information is included under the Institutional Review Board Statement.

-        L107-108: Did you collect the samples from live or dead animals? Give more details about the animals (e.g., age, breed, etc.).

Serum was collected from live animals and tissues from dead animals. Pigs belonged to mixed breeds and at different ages. This information was added to the manuscript.   

-        L157-170: Add appropriate references. Added

-        L213-220: Add a table with the used scoring system, giving details per score 0, 1, 2, and 3 and adding the appropriate reference.

A reference for the used scoring system was given - . Mittelholzer1 C, Moser2 C, Tratschin JD, Hofmann MA. Analysis of classical swine fever virus replication ki-netics allows differentiation of highly virulent from avirulent strains. Vet Microbiol. 2000 Jun 12;74(4):293-308.

The details of each score and a table is in the reference, and therefore we believe such a table will not add extra value to this manuscript.

Discussion 

  •        You could include a paragraph of thoughts on the economic impact of your results on the swine industry in South Asia.   A sentence describing possible impact of genotype 2.6 in the control of CSF in Asia was added.  

Minor comments

-        L21:.. the full-length. Corrected

-        L36: .. Paraguay, Uruguay, and some areas Corrected

-        L52: .. into high, moderate, and low. Corrected

-        L79: .. because of trans-placental transmission Changed

-        L129: … in each homogenizing. Corrected

-        L130: .. 10-second runs with 10-second pauses Corrected

-        L135: … homogenate were used for. Changed

-        L260: .. during the lactation phase….Not clear. We  believe that this comment is not relevant to this manuscript

-        L159: .. CSFV-infected PK-15 cell. Corrected

-        L212: .. clinical findings of each animal. Corrected

-        L223: .. On each sampling day…Corrected

-        L581: One-third of those farms.. Corrected

-        L582: More than half of the pigs.. Corrected

-        L609: .. 6-week-old weaned piglets. Corrected

-        L612: The fever remained above. Corrected

-        L613: Despite the fever. Corrected

-        L633: .. the FAO-led CSF eradication. Corrected

-        L641: .. the Northern departments. Corrected

-     L643: .. sequences suggests possible. Corrected

Round 2

Reviewer 2 Report

Comments and Suggestions for Authors

This paper is a study that has researched CSFV occurring in Colombia from various angles over several years. For this paper to be finally published, the following minor content needs to be revised. when writing the source of the materials, after the city or state and country has been marked once, you only need to write the company name from the second time onwards.